# Exploring Optimal Biomarker Sources: A Comparative Analysis of Exosomes and Whole Plasma in Fasting and Non-Fasting Conditions for Liquid Biopsy Applications

**DOI:** 10.3390/ijms25010371

**Published:** 2023-12-27

**Authors:** Masaki Nasu, Vedbar S. Khadka, Mayumi Jijiwa, Ken Kobayashi, Youping Deng

**Affiliations:** Department of Quantitative Health Sciences, John A. Burns School of Medicine, University of Hawaii, 651 Ilalo Street, Honolulu, HI 96813, USA; vedbar@hawaii.edu (V.S.K.); jijiwa@hawaii.edu (M.J.); kendk@hawaii.edu (K.K.)

**Keywords:** liquid biopsy, fasting, exosomes, sncRNA, metabolites

## Abstract

The study of liquid biopsy with plasma samples is being conducted to identify biomarkers for clinical use. Exosomes, containing nucleic acids and metabolites, have emerged as possible sources for biomarkers. To evaluate the effectiveness of exosomes over plasma, we analyzed the small non-coding RNAs (sncRNAs) and metabolites extracted from exosomes in comparison to those directly extracted from whole plasma under both fasting and non-fasting conditions. We found that sncRNA profiles were not affected by fasting in either exosome or plasma samples. Our results showed that exosomal sncRNAs were found to have more consistent profiles. The plasma miRNA profiles contained high concentrations of cell-derived miRNAs that were likely due to hemolysis. We determined that certain metabolites in whole plasma exhibited noteworthy concentration shifts in relation to fasting status, while others did not. Here, we propose that (1) fasting is not required for a liquid biopsy study that involves both sncRNA and metabolomic profiling, as long as metabolites that are not influenced by fasting status are selected, and (2) the utilization of exosomal RNAs promotes robust and consistent findings in plasma samples, mitigating the impact of batch effects derived from hemolysis. These findings advance the optimization of liquid biopsy methodologies for clinical applications.

## 1. Introduction

Liquid biopsy, which is preferentially performed on circulating nucleic acids extracted from blood plasma, can be a tool to overcome limitations faced by conventional tissue biopsies [1,2,3]. In plasma samples, biological elements such as circulating cell-free DNA, mRNA, non-coding RNA, proteins, metabolites, and lipids have been detected [1,3,4,5]. Among these biological materials used in liquid biopsies, circulating cell-free micro-RNAs (miRNAs) have been studied as promising cancer biomarkers [6,7,8,9]. Many cancer studies correlate the presence of specific miRNAs in plasma with different types of cancer. However, there are discrepancies in the reported miRNA species among studies of plasma biomarkers. Recent reviews have reported tremendous heterogeneity and inconsistency in cancer-related plasma miRNA markers [7,8,9,10]. Although many miRNAs have either shown conflicting results or have been reported in a single study, few miRNA biomarkers have been validated in multiple independent studies [7,8,9,11,12]. This is due to a lack of standardization in the source of circulating miRNAs (serum/plasma/urine/saliva/etc.), isolation protocol, analytical approach (e.g., using microarray versus RT-qPCR), or sample size. Most of the studies focus on using plasma as the miRNA source. Plasma that is separated from blood samples is supposed to be cell-free, and thus any contamination in plasma with cellular RNAs is supposed to be reported [13]. Recent studies have shown that hemolysis (the rupture of red blood cells) occurring during blood collection or sample processing can cause the release of intracellular biological materials into plasma specimens [14,15,16]. Hemolysis is a very common cause of error in the clinical laboratory, accounting for 40% to 70% of unsuitable samples [14,16]. Release of intracellular components such as erythrocytes and platelets upon hemolysis can lead to an increase in the concentration of intracellular analytes. Recent studies have shown that the frequency of hemolysis is highly variable among institutions because it is influenced by the experience level of the phlebotomist and the training of personnel obtaining the specimens [15,16,17,18,19]. Therefore, the degree of hemolysis among labs and clinics can cause a difference in the quality of plasma.

New studies have revealed that cancer-derived exosomes (extracellular vesicles, or EVs) are enriched with sncRNAs signatures of their cell of origin [9,20,21]. The analysis of exosomal miRNAs is superior to whole plasma analysis because exosomes contain more meaningful biomarkers compared to miRNAs in whole plasma [1,9,13,20]. Exosomes are endosomally derived small membrane vesicles (50–100 nm) which are released from cells into the extracellular environment, such as plasma. Recently, many studies have shown that exosomes contain proteins, lipids, metabolites, functional mRNA, sncRNAs, and other non-coding RNAs (such as long non-coding RNAs (lncRNAs)) [1,2,13,22,23]. Exosomes act as a cargo, transferring these biologically active components into targeted cells, playing the role of a cell-to-cell communicator [2,13,20,24]. The content of exosomes, such as mRNA and miRNA, is dependent on the cell of origin and can be transferred into adjacent or distant cells [2,23,25,26]. This suggests that exosomes contain highly specific RNA biomarkers. The membrane of the exosome protects the content RNAs from degradation in the bloodstream (Figure 1A). The RNA content in the exosome is thus relatively stable compared to cell-free RNA floating in plasma [9,20,27]. These advantages suggest that the investigation of exosomal RNAs could provide consistent data to produce reliable cancer biomarkers compared to those from whole plasma RNA contaminated unnecessary with cellular RNAs. Exosomes are therefore an attractive source for diagnostic and prognostic biomarkers [9,20,27,28,29,30,31]. We hypothesize that the exosomal RNA is the most consistent and reliable source of cancer biomarker for precision medicine. Further investigation of the comparison between exosomal RNAs and RNAs from whole plasma is required.

Recently, metabolites, including lipids, are brought to attention as liquid biopsy biomarkers [32,33,34]. Aberrant metabolism, including lipid metabolisms and fatty acid metabolism, have been reported in cancers and are knowns to be one of the hallmarks of cancer [35,36]. However, the findings in cancer metabolites studies are extremely varied, depending on the application of study designs, methods, and analyses [32,36]. Our hypothesis is that the difference in fasting status during these studies is the cause of difficulties in metabolite biomarker studies. We would like to identify reliable metabolites which are not affected by fasting/non-fasting status. Very few papers from metabolite biomarker studies mention whether fasting status was taken into consideration or not. Because fasting has previously been the standard before blood sampling for a lipid profile in most countries, lipid profiles are conventionally measured in the fasting state [37]. However, in recent times, the shift toward using non-fasting rather than fasting lipid profiles has been increasingly debated [38,39]. It has been proposed that after normal food intake, concentrations of lipids, lipoproteins, and apolipoproteins differ only minimally from the fasting to the non-fasting state in the general population [37,40].

In this project, we would like to prove that the contamination of cell-free RNA profiles with degraded cellular RNAs is the main cause of the discrepancy and inconsistency in blood plasma biomarker studies (Figure 1A). We hypothesize that the biological materials in exosomes were not contaminated by other materials in blood. The other hypothesis is that the contents of exosomes are not affected by fasting/non-fasting status compared to other cell-free components in whole plasma. Therefore, exosomes are the better source for a biomarker study compared to the biological materials found in whole plasma. To prove these hypotheses, we compared the RNA profiles in the exosome with the RNA profiles in whole plasma. We analyzed small RNA expression levels in plasma and exosome samples from four healthy individuals using small RNA-Seq based on NGS technology. We also compared fasting profiles and non-fasting profiles (Figure 1B–D). We collected blood samples from participants immediately following overnight fasting and again after breakfast (non-fasting sample). Our data showed clearly that RNA content is not affected by fasting and non-fasting status. We also showed the differences between exosomal RNA and whole plasma RNA. Our data suggested that the exosomal RNA is not contaminated with cellular blood RNA. In contrast, the whole plasma RNA can be contaminated with cellular blood RNA. Therefore, the exosomal RNA is suitable for cancer biomarkers to avoid batch effects. We had a difficult time comparing the differences in metabolites between plasma and exosome. However, we found some metabolites which are not influenced by fasting/non-fasting status in plasma.

## 2. Results

### 2.1. RNA Profile Is Not Affected by Fasting/Non-Fasting Status

Exosomal RNA samples showed the presence of 2400 miRNA species, 68 piRNAs, 233 tRNAs, and 85 other small RNAs, including snoRNAs. In contrast, whole plasma RNA samples contained 2425 miRNAs, 92 piRNAs, 252 tRNAs, and 96 other small RNAs. Correlation analysis showed that fasting/non-fasting status did not substantially affect RNA profiles, with a high correlation coefficient of 0.9994 for exosomal small RNAs and a slightly lower coefficient of 0.9868 for plasma small RNAs (Figure 2A,B). These findings highlight that fasting is not a prerequisite for discovering RNA biomarkers, either from plasma or exosome. Notably, exosomal miRNA profiles showed the correlation coefficient of 1.00 (Figure 2C). Although the stability of exosomal profiles was higher than that of plasma profiles, the overall difference was marginally significant (*p* = 0.0618679). Correlation coefficients for piRNA and tRNA in exosomes also exceeded those in plasma (Figure 2D,E), while other small RNA samples show strong positive correlations (Figure 2F).

### 2.2. Exosomal RNA Profile Is Different from Whole Plasma RNA Profile

The principal component analysis (PCA) plot, utilizing normalized RNA-seq data, uncovered marked differences between exosomal and plasma RNA profiles (Figure 3A). While exosomal RNA profiles demonstrated remarkable consistency among different healthy individuals (blue dots are hidden by orange dots), plasma RNA profiles showed considerable heterogeneity, with the dots scatted around the plot. This observation indicates that the expression levels of exosomal small RNA remain stable regardless of whether the individual is fasting or not, thereby implying their stability compared to whole plasma RNAs. Interestingly, some important RNA variants, such as miR-486-5p, miR-92a-3p, has-miR-16-5p, and has-miR-451a, which are known to be enriched in blood cells, contributed to the heterogeneity in plasma RNA profiles (Figure 3B). Plasma RNA profiles have this heterogeneity because concentrations of these blood cell miRNAs are different among different individuals.

To delve deeper, we evaluated the expression level of 28 miRNAs which are reported to derived from platelets and erythrocytes, including has-miR-1-3p, miR-10a-5p, miR-10b-5p, miR-126-3p, miR-144-3p, miR-144-5p, miR-145, miR-150-5p, miR-16-5p, miR-183-5p, miR191-5p, miR-197-3p, miR-21-5p, miR-223-3p, miR-23a-3p, miR-24-3p, miR-26a-1-3p, miR-27b-3p, miR-28-3p, miR-29b-3p, miR-320a, miR-33a-5p, miR-411-5p, miR-423-3p, miR-423-5p, miR-451a, miR-486-5p, and miR-92a-3p [41,42,43,44]. Most of these miRNAs exhibited higher expression in whole plasma RNA samples than in exosomes (Figure 4A). The heat map was generated using log2 fold change values normalized by average of all DEseq2 read. Out of 28 miRNA, 25 showed significant difference with a *p* value lower than 0.01 while has-miR-411-5p had a *p* value of 0.0124. This result explains that blood cell-derived miRNAs can introduce bias in the measurement of miRNA biomarkers’ expression levels in whole plasma RNA profiles. (Figure 1A). Expression levels of small RNAs were compared between plasma and exosomes based on their types, including miRNA, piRNA, tRNA, and other small RNAs, including snoRNA (Figure 4B–E). The heat map for miRNAs was constructed using the top 18 upregulated and 18 downregulated miRNAs in exosome compared to plasma log2 fold changes (Figure 4B). A total of 810 miRNAs exhibited significant differences in expression between exosomal RNAs and plasma RNAs (*p* < 0.05), confirming distinct miRNA profiles in exosomes and plasma. Among these, 28 out of 36 miRNA showed significant changes in fold change (*p* value < 0.01), with 5 miRNAs showing *p* values between 0.04 and 0.01. Notably, has-miR-6731-5p, the most upregulated miRNA in exosomes, had an average NGS normalized read of 465.77 in exosome, compared to 22.11 in plasma. Conversely, the most downregulated miRNA in exosomes, has-miR-192-5p, had 13.46 read in exosome and 1853.93 in plasma. Additionally, 7 miRNAs (has-miR-10a-5p, has-miR-10b-5p, has-miR-126-3p, has-miR-144-3p, has-miR-144-5p, has-miR-451a, has-miR-486-5p) were identified as top upregulated miRNAs in plasma and were reported to have originated from blood (Figure 4A). In the case of piRNAs, the expression level of 30 piRNAs was significantly different between exosomal piRNAs and plasma piRNAs (*p* < 0.05). The heat map featured the top 11 upregulated and 11 downregulated piRNA in exosomes (Figure 4C). Interestingly, some of these piRNA, such as hsa_piR_018573, hsa_piR_001318, hsa_piR_019324, and hsa_piR_018849 have been reported as downregulated piRNAs in prostate cancer urine [45], suggesting their potential as consistent exosomal caner biomarkers. Similarly, tRNA expression levels showed significant difference between exosomal tRNAs and plasma tRNAs, with the top18 upregulated and 18 downregulated tRNA forming distinct patterns (Figure 4D). For other small RNAs (snoRNA), the heat map revealed significant differences in the expression level of 59 small RNAs between exosome and plasma (*p* < 0.05). The top 12 upregulated and 12 downregulated small RNAs formed distinct clusters (Figure 4E). It is noteworthy that only limited research has focused on these other small RNAs in exosomes. Nevertheless, the detection of ACA31 was found to be increased in EVs released from cells which were infected with West Nile virus [46].

### 2.3. Unstable Metabolites in the Plasma Profiles, Which Is More Reliable than the Exosome for Metabolomics Study

Exosomal metabolites showed limited detection, with only 261 out of 513 total metabolites found in whole plasma samples (Appendix A). Notably, lipidomic type species (fatty acids, TG-DG, lipids) were detected as being relatively more prevalent than amino acids and bile acids in exosome. Both the correlation assay and the volcano plot assay indicated that exosome metabolites were more affected by fasting status than plasma (Figure 5 and Figure 6 and Appendix A). Additionally, the concentrations of exosome metabolites were too low, mostly in the single or double-digit pmol/mL range, to yield a significant outcome. Thus, plasma metabolite profiles were primarily utilized to identify stable or unstable metabolites between fasting and non-fasting conditions. Both plasma and exosome metabolites profiles responded to the fasting, non-fasting status (Figure 5A,B), with exosomes exhibiting a higher sensitivity to these conditions compared to whole plasma. The correlation coefficient was 0.9435 in exosome, and slightly higher at 0.9634 in whole plasma. Additionally, correlation analysis between fasting and non-fasting in plasma revealed that bile acids were relatively unstable compared to other type of metabolites (Figure 5C–F). These findings emphasize the significance of choosing the suitable metabolite species for metabolomics research.

We have identified metabolite species which were significantly affected by fasting/non-fasting status in whole plasma profiles (Figure 6 and Appendix A). Among 30 amino acids species examined, only beta alanine showed a slight increase following food consumption (log2 FC = 1.055, *p* value = 0.0027) (Figure 6A and Appendix A). The log2 fold changes of the remaining 28 amino acid species ranged from 0.37 to −0.27, indicative of minimal variations. The concentration of 10 out of 35 bile acids were increased with log2FC > 1, while 7 bile acids experienced reduction with log2FC< 0.48. Notably, bile acid such as cholic acid (NorCA), glycochenodeoxycholate (GCDCA) and glycodeoxycholic acid (GDCA) significantly increased post-food consumption, boating log2FC values of 1.62, 1.59, and 1.41 respectively (Figure 6B). On the other hand, the concentrations of deoxycholic acid (DCA), and allolithocholic acid (alloLCA) were significantly decreased with log2 FC of −0.73, and −0.89, respectively (Figure 6B). The concentration of most fatty acids (24 out of 39) decreased after food consumption (Figure 6C). However, only 2 fatty acids, caprylic acid (C8:0) and hexanoic acid (C6:0), significantly increased in concentration with log2FC of 0.87(*p* value = 0.0046), and 0.69 (*p* value = 0.014), respectively (Appendix A). This phenomenon, fatty acids concentration rising during fasting and subsequently decreasing after refeeding, is consistent with findings reported by Whitehurstn et al. [47]. Our study also revealed a significant decrease in various fatty acids, including arachidonic acid (C20:4(cis_5,8,11,14)), eicosapentaenoic acid (EPA) (C20:5(cis_5,8,11,14,17)), adrenic acid (docosatetraenoic acid, C22:4(cis_7,10,13,16)), and docosahexaenoic acid (C22:6(cis_4,7,10,13,16,19)) (Figure 6C and Appendix A). Notably, caprylic acid (C8:0) showed the most significant change (*p* value = 0.0046), increasing after refeeding.

Out of 329 lipid species examined the majority (296) showed no significant changes (Figure 6D), indicating that only a (10%) of lipid species were influenced by fasting. Log2 fold change for 281 lipids fell within the range of −0.5 to 0.5. However, 20 lipid species including 12 phosphatidylethanolamines (PE), 4 lyso Phosphatidylethanolamines (LPE), 2 ceramides (Cer), and 2 phosphatidylserines (PS), exhibited significant upregulation (Appendix A). Additionally, 13 lipids species such as C8 Octanoylcarnitine, C10 Decanoylcarnitine, C12 dodecanoylcarnitine, C12:1, C14:1 (cis-5-Tetradecenoylcarnitine), C14:1-OH, C14:2, C16:2, C18:1-OH. C14:1 and C14:2 were reported to be elevated during fasting [48,49]. These findings suggest that the significantly altered lipids may not be appropriate as cancer lipid biomarkers. The volcano plot analysis of exosome profiles showed that only certain TG-DG and lipid species had significant up- or downregulation between fasting and non-fasting states. The profiles of exosomal TG-DG exhibited a similar trend to that of the plasma profiles. (Appendix A). Almost all TG-DGs were upregulated after food consumption. Exosomal lipid species showed 5 significantly altered lipids, 4 of which were also found in plasma lipid profile: C8, C10:1, C10 downregulated, PE(36:2) upregulated (Appendix A). There were no significant changes in exosomal fatty acids, amino acids, or bile acids found via volcano assay.

## 3. Discussion

In this study, there were some pressing questions that needed to be answered. Firstly, we explored the necessity of fasting in liquid biopsy biomarker studies, and our findings indicate that fasting does not significantly impact exosomal small RNA profiles. Secondly, our study aimed to identify the superior source of cancer biomarkers: purified exosomes or whole plasma. Our analysis of whole plasma RNAs revealed the presence of exosomal RNA, cell-free RNAs, and potentially cellular RNAs. Crucially, it was conclusively demonstrated that the profiles of exosomal RNA and whole plasma RNA from the same blood samples differ when their differences were compared. These findings support the idea that diverse levels of cellular RNA contamination may lead to varying degrees of background noise in whole plasma RNA samples. Consequently, we advocate for exosomal RNA profiles as they exhibit fewer batch effects. Since exosomes originate from viable tumor cells, exosomal RNAs provide a promising approach for identifying specific biomarkers that reflect broader physiological states and diseases, including cancer. Plasma samples were utilized instead of serum in this study. Several studies have compared plasma and serum as sources of biomarkers [10,50]. These studies demonstrated that there are slight differences between the profiles of serum and plasma. However, it does not provide a clear conclusion as to which one is superior [7]. In terms of miRNA profiles, serum samples were found to have a higher concentration but a lower number of different miRNA species detected [51,52]. It was proposed that the coagulation process with serum preparation could impact the number of circulating miRNA species. Another study revealed that serum samples contained a higher quantity of platelet-derived EVs than plasma [53]. This is also due to coagulation reactions, which cause the release of EVs from platelets and can alter the profiles of EVs. The presence of platelet-derived EVs in serum may compromise the accuracy of cancer biomarker studies. These studies support the idea that focusing on EVs purified from plasma, rather than whole plasma or serum, is the first step in reducing contamination and background information for RNA biomarker studies.

Compared to RNA profiles, metabolites were affected more by fasting/non-fasting status. The extent to which the metabolites were affected by fasting status had variable results, with different metabolite species yielding different profiles. The effect of fasting on lipid species was minimal; 90% of lipid species did not show significant changes. These results suggest that fasting is not necessary for plasma biomarker studies, but it is important to choose lipid species thar are not affected by fasting. The concentration of some metabolites in plasma can be changed by fasting itself [48,54,55,56]. Studies have shown that 20% to 50% of metabolites can be affected by fasting/non-fasting status [55,57]. For some metabolites, the effects of fasting status are significant, although for most metabolites the effects are modest. The results from our study can also support previous studies. For example, fatty acids such as palmitic acid (C16:0) are proposed to regulate metastasis [58], and they have a relatively stable baseline in our data (FC 0.8088, Log 2 FC −0.30614, *p* value 0.1131). This suggests that palmitic acid could be a good biomarker. One NSCLC study identified that LysoPC 20:3 (lysoPC a C20:3 in our lipid data), PC ae C40:6, and citric acid (amino acids) were significantly different between healthy controls and stage I/II NSCLC [34]. These are reliable biomarkers since the effect of fasting/non-fasting is very small in our data; LysoPC 20:3 (fold change 0.9245, log2FC −0.11312, *p* value 0.06), PC ae C40:6 (fold change 1.0607, log2FC 0.085, *p* value 0.71), and citric acid (amino acids) (fold change 1.08, log2FC 0.114, *p* value 0.41). Our data revealed that bile acids and fatty acids may not be a good candidate for biomarker studies. The concentration of these metabolites decreased significantly after food consumption. This effect in which the concentration of metabolites increased during fasting and decreased after refeeding is observed in Whitehurstn et al. [47]. It is reported that fasting can increases the concentration of free fatty acid and glycerol in blood plasmas suggesting the abnormal condition during fasting [47]. Fasting itself may cause drastic effects on these metabolite biomarkers. Fold change analysis alone revealed that TG, DG fatty acids and bile acids were unstable (48.72%, 48.57%) (Appendix A). 19 out of 39 fatty acids, 28 out of 30 amino acids and 292 out of 329 lipids were within the fold change threshold. Out of 35 bile acid species, 17 showed drastic changes (10 increased and 7 decreased). A similar report was found in a study by Townsend et al. [59]. Bile acid had the most variability among other metabolites such as amino acid derivatives, amines, amino acids, lipids and lipid metabolites [60]. Therefore, bile acids may not be appropriate to use as a liquid biopsy biomarker. Some advantages to using non-fasting samples rather than fasting samples are proposed: patients do not need to fast, and clinicians can have the lipid profile determined at any random time of day [37]. Fasting can be a barrier to population screening. It is unpopular with children, and it is often unsuitable for patients with diabetes [61]. It is difficult to conduct a large-scale biospecimen collection when participants are spread across a wide geographic area. The standardization of fasting and blood collection protocol is difficult to achieve. For instance, some patients may be unable to adhere to the rigorous fasting protocols necessary for a blood draw [59,62]. Therefore, we conclude that fasting is not necessary, but it is important to use metabolite species with stable baseline concentration regardless of fasting status.

Our study encountered two notable limitations. First, our results suggested that 500 μL of plasma to isolate EVs was adequate for RNA profiles but was not sufficient to isolate enough EVs for metabolite study. It was not possible to obtain enough data to discuss the effect of fasting on exosomal metabolites. It was difficult to detect enough metabolites to make a conclusion for exosomal metabolites. It is also important to discuss the fact that no single standardized method exists yet for the perfect isolation of exosomes for their use in cancer biomarker studies [63,64,65]. There is still some debate as to whether it is even possible to isolate the intact and pure exosomes. The ultracentrifugation-based technique was initially the commonly used method for isolating EVs. This technique does have its limitations; it is time-consuming and requires specialized equipment that is not available at clinical labs. This can therefore pose significant challenges for their use in clinical diagnostic tests [66]. Another difficulty is the optimization of centrifuge speed and time: the excessive spinning results in damage of EVs, and inadequate centrifugation may result in a high level of impurities [65]. In our study we used a spin column kit to separate any large particles such as membrane fragments, apoptotic bodies, and other cell fragments from small EVs (which have the size range between 30 and 100 nm). This kit also utilizes another membrane with an immobilized lectin-based compound to capture the glycosylated EVs [67]. The presence of glycoconjugates is involved in EV biogenesis, and in the efficient uptake of EVs by recipient cells [67]. This Capturem Extracellular Vesicle Isolation Kit (Mini) consists of these different spinnable membrane filter columns to capture EVs. This provides a unique solution for isolating concentrated, high-purity EVs in the range of 30–100 nm. (Appendix A). The data that were collected using this kit were satisfactory to examine exosomal RNA profiles. However, we understand that it is still challenging, and there is debate about exosome (extracellular vesicles, or EVs) research [29,68,69]. Current isolation techniques may still contain some other vesicles and lipoprotein contaminant [64]. The detailed biogenesis of exosomes and their basic functions and the lack of specific biomarkers are still under investigation [69]. Standardized isolation and characterization methods must be developed. The ultimate goal is to identify exosomes released directly from cancer cells and to evaluate the content of exosomes to determine specific personalized cancer therapy.

Secondly, our sample size was small. It was suggested that metabolite liquid biopsy needs a larger number of participants, as a large sample size is critical for reaching any conclusion with moderate confidence [56]. Regarding RNA profiles, we observed consistent effects between individuals of fasting/non-fasting status, despite the small sample size. This observation was explained with the PCA plot (Figure 3). However, it is important to note that this study would benefit more from a larger sample size.

## 4. Materials and Methods

### 4.1. Plasma Samples and Whole Plasma RNA Purification

We used blood samples from 4 healthy volunteers to understand the baseline expression level of biomarkers (Appendix A). These donors gave written informed consent for the use of their blood samples in accordance with IRB protocol number 2021-01033. The volunteers underwent an overnight fast and were subsequently subjected to a blood test (referred to as “fasting samples”). After the fasting blood samples were collected, the participants ate a nutritious breakfast. Following this, they were instructed to avoid any food or drinks except water. Additional blood samples were collected from the individuals three hours and five hours after the breakfast (termed “non-fasting samples”). Blood samples were collected into BD K2 EDTA tubes. Plasma samples were isolated right after the blood collections via centrifuging at 1500× *g* for 15 min. Total RNA, including small RNA, was extracted from 200 μL of whole plasma using miRNeasy Serum/Plasma Kit (Cat. No.217184, Qiagen, Hilden, Germany), following the manufacture protocol.

### 4.2. Exosome Preparation and RNA Samples

Exosomes were purified from 500 μL of plasma by using Capturem™ Extracellular Vesicle Isolation Kit (Cat. No. 635741, Takara Bio USA, Inc., San Jose, CA, USA) Exosomes were eluted with 200 μL elution buffer. Total RNA, including small RNA, was subsequently extracted from these exosome samples using the miRNeasy Serum/Plasma Kit (Qiagen, Hilden, Germany).

### 4.3. Small RNA Sequencing

Library preparation and small RNA sequencing were conducted by the Genomics and Bioinformatics Shared Resources (GBSR) at the University of Hawaii Cancer Center (UHCC). The construction of the small RNA sequencing library employed the QIAseq miRNA Library Kit (Cat. No.331505, Qiagen, Germany) and the QIAseq miRNA NGS 12 Index IL (Cat. No.331582, Qiagen, Germany) according to the manufacturer’s instructions. The quality of libraries was validated on the Agilent 2100 Bionanalyzer using a high-sensitivity DNA chip. The next-generation sequencings were carried out on the Illumina NextSeq 500 platform. The analysis of sequencing data was conducted by GeneGlobe (QIAGEN). The sequencing data were normalized using the DESeq2 method to evaluate RNA expression level.

### 4.4. Metabolic/Lipidomic Analysis

Metabolite and lipid concentration analysis were carried out using ultra-performance liquid chromatography (LC) coupled with tandem mass spectrometry (MS) at the UHCC Metabolomics Shared Resource. The normalization standards and stable isotope-labeled standards were used for the measurements. We checked a total of 513 metabolites, including 329 Lipid species, 35 bile acids (BAs), 55 triacylglycerol (TGs), 25 diacylglycerol (DGs), 39 fatty acids, and 30 amino acids. The lipidomic and metabolomic concentration measurement were conducted following the protocol published in reference [70]. To distinguish between fasting/non-fasting states, we focused on TGs and DGs. These species are well known to show clear increase when people eat after overnight fasting [39,61]. Out of 80 TG and DG species, 60 were significantly increased after eating (Figure 1D). As an example, the concentration of Triglyceride (TG 56:2) exhibit a substantial 4.32-fold change three hours after the breakfast, with two-sided *p* value of 0.017 (Figure 1C). Building upon these findings, we explored the changes in other metabolites and small RNA species. Statistical analyses were carried out using Prism 9 (Graphpad Software Inc., LA Jolla, CA, USA), and MetaboAnalyst (McGill University, Montreal, CA, USA). We duplicated samples from each volunteer, resulting in a total of 8 experiments for this metabolites experiments.

## 5. Conclusions

In conclusion, we propose that exosomes (EVs) serve as a superior sources of nucleic acid biomarkers, and we challenge the necessity of fasting in the investigation of RNA and metabolite biomarkers in liquid biopsy studies with plasma. Our study clearly showed that exosomal RNA profiles are different from whole plasma RNA. The importance of utilizing exosomal RNA as a strategy to mitigate batch effects in RNA biomarker studies is highlighted by the potential contamination of whole plasma RNAs with cellular RNAs. Regarding metabolites, our research indicates that it is crucial to carefully select metabolites that are not affected by fasting or non-fasting in order to guarantee the accuracy of biomarkers. Fasting itself may have drastic effects on certain metabolite biomarkers. The findings of our study indicate that TG, DG, fatty acids, and bile acids are inadequate biomarkers, whereas amino acids and lipid species showcase better stability for biomarker applications. Our data suggests that conducting plasma analysis for lipidomics and metabolomics is feasible without imposing fasting requirements. However, clear limitations persist, necessitating the avoidance of metabolites influenced by food consumption. This research highlights the potential to enhance the precision and practicality of liquid biopsy studies for cancer biomarker research. To establish reliable and clinically consistent biomarkers, especially for RNA biomarkers, it is suggested that the first step should be to standardize the use of exosomes.

## Figures and Tables

**Figure 1 ijms-25-00371-f001:**
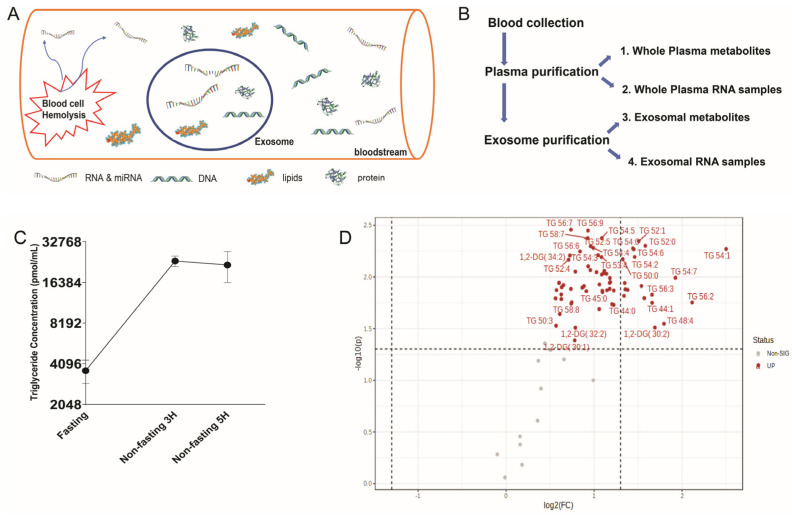
Hypotheses and experimental designs. (**A**) Exosomal RNAs are protected from bloodstream environment. Cell-free RNAs in blood are not only derived from exosomes but also from hemolysis of blood cells. The hypothesis of this study is that blood cell RNAs resulting from hemolysis can lead to batch effects in RNA expression levels across various cohorts. (**B**) Experimental scheme. We compared biological material in exosomal samples and whole plasma samples: Plasma samples were used to purify (1) whole plasma metabolites, (2) whole plasma RNAs and exosomes. Exosomes were used to further purify (3) exosomal metabolites and (4) exosomal RNAs. (**C**) Blood collections were conducted at three different time points to compare the profiles of individuals while fasting and not fasting. (1) Following an overnight fast. (2) Three hours post-breakfast. (3) Five hours after breakfast. The graph displays the fluctuation in concentration of triglyceride (TG 56:2) over time. (**D**) A volcano plot was created, using a fold change threshold of 1.4 and a t-tests threshold of 0.05, to demonstrate the impact of fasting/non-fasting status. Triglyceride (TG) and diglyceride (DG) concentration at non-fasting were normalized by fasting status. Red dots indicate significantly increased TG and DG 3 h after consuming breakfast. Out of 80 species, 68 exhibited a significantly increased concentration.

**Figure 2 ijms-25-00371-f002:**
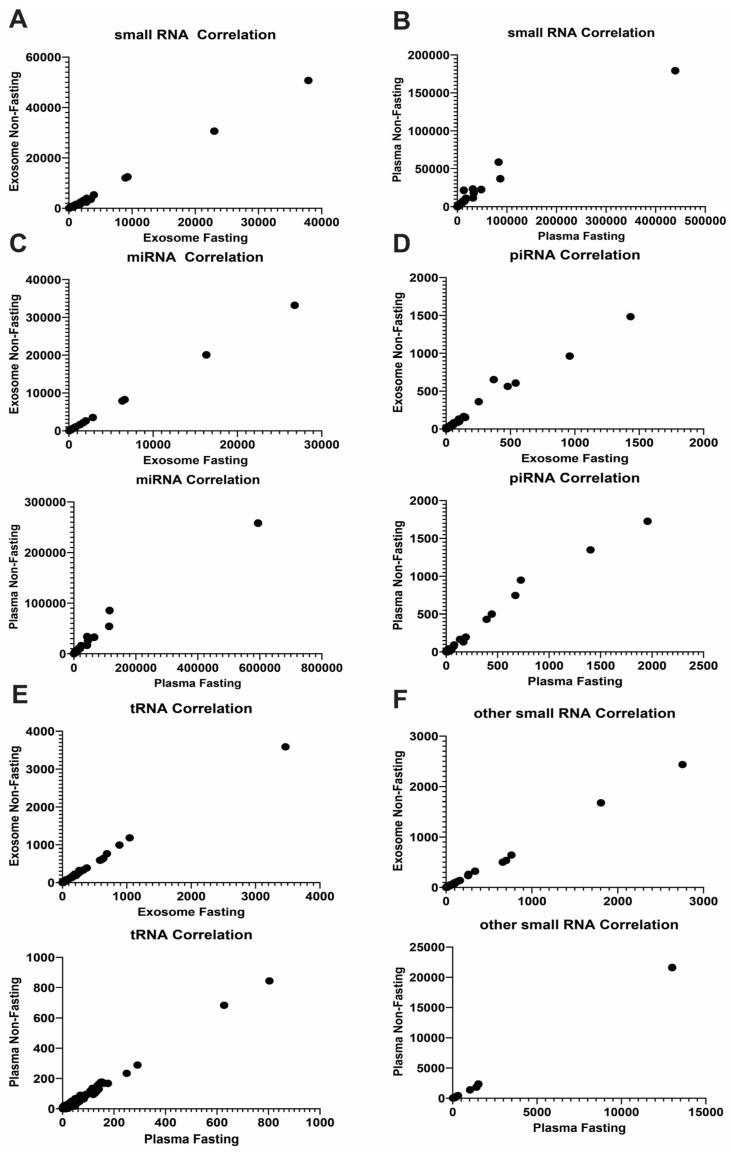
Correlation assay to compare differences between fasting and non-fasting RNA profiles. (**A**) The correlation coefficient for exosomal small RNAs is 0.9994. (**B**) The correlation coefficient r for small RNAs in whole plasma is 0.9868. Correlation assay in different type of small RNAs (**C**–**F**). (**C**) miRNA, Exosome r = 0.9999, whole plasma r = 0.9903. (**D**) piRNA, Exosome r = 0.9896, whole plasma r = 0.9825. (**E**) tRNA, Exosome r = 0.9992, whole plasma r = 0.9889. (**F**) Other small RNA, Exosome r = 0.9987, whole plasma r = 0.9996.

**Figure 3 ijms-25-00371-f003:**
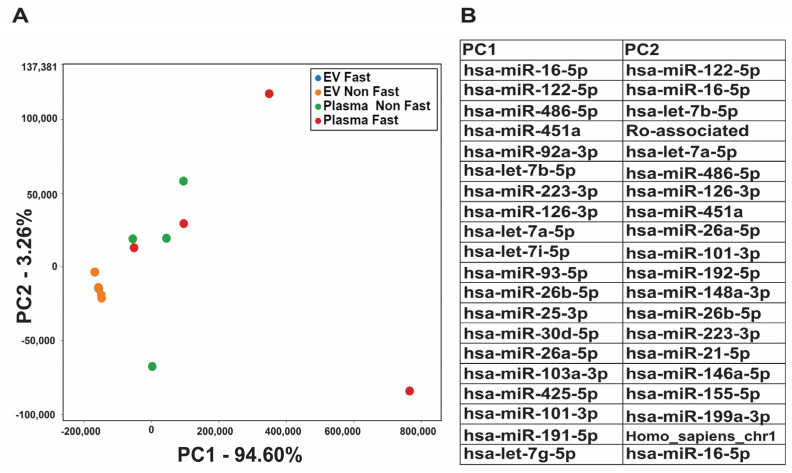
Principal component analysis (PCA) to compare differences between exosomal RNA profiles and whole plasma RNA profile. (**A**) PCA plot comparing exosomal (EV) RNAs at fasting (blue), exosomal (EV) RNAs at non-fasting (orange), plasma RNAs at fasting (red), and plasma RNAs at non-fasting (Green). PC1 captures the most important variation, PC2 captures the second most important variation. (**B**) Top 20 small RNA species in PC1 and PC2. These RNAs were differentially expressed with high variance between these two profiles.

**Figure 4 ijms-25-00371-f004:**
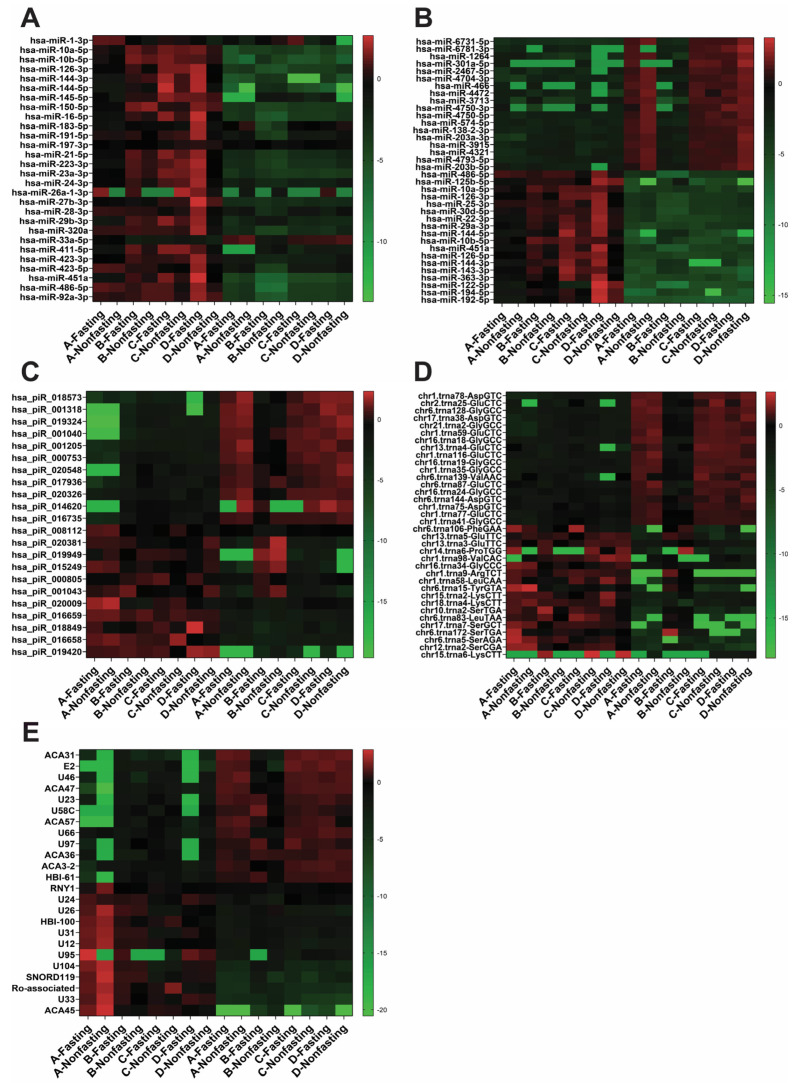
Heat maps showing differences between exosomal RNA profiles and whole plasma RNA profiles. (**A**) Heat map to compare expression level of platelet- and erythrocyte-derived miRNAs. Highly expressed miRNAs in blood cells are also highly expressed in plasma, but not in exosomes. Heat maps display the differentially expressed small RNA in miRNA (**B**), piRNA (**C**), tRNA (**D**), and other RNAs including snoRNA (**E**) between exosomal and whole plasma profiles.

**Figure 5 ijms-25-00371-f005:**
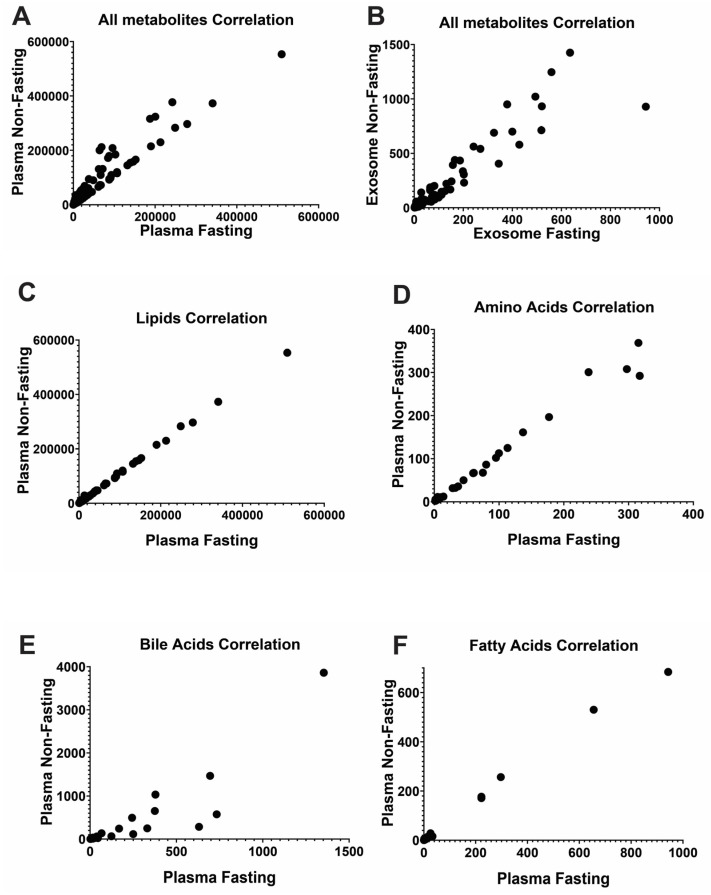
Correlation assay comparing metabolite profiles between fasting and non-fasting conditions. (**A**) Correlation between fasting and non-fasting for all plasma metabolites. r: Correlation coefficient. r = 0.9634. (**B**) For all metabolites in exosome r = 0.9435. (**C**) Only for lipid species in plasma r = 0.9996. (**D**) Only for amino acids r = 0.9911. (**E**) For bile acids r = 0.8936. (**F**) For fatty acids r = 0.9979.

**Figure 6 ijms-25-00371-f006:**
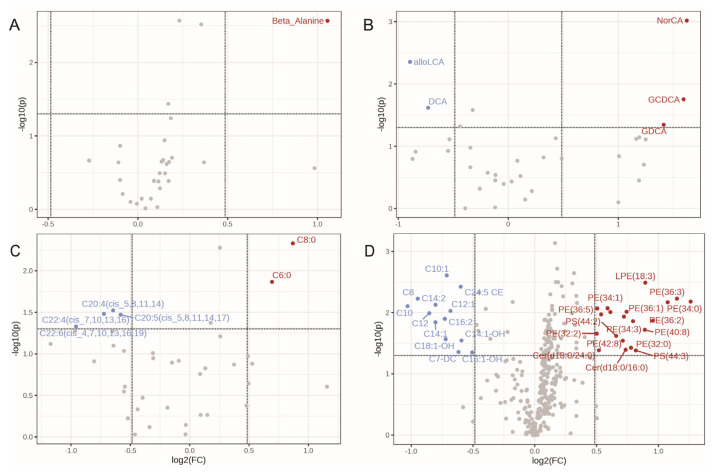
Volcano plots display the metabolite species with significant changes after consuming food. Fold change threshold of 1.4 (*x* axis) and *t*-test thresholds of 0.05 (y) were utilized. Red dots: upregulation; blue: downregulation; gray: non-significant changes. (**A**) Amino acids. (**B**) Bile acids. (**C**) Fatty acids. (**D**) Lipids.

## Data Availability

RNA sequence data are deposited at NCBI-GEO under accession number GSE216556. Please contact the authors for further data requests.

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
