# Peer review of "Exploring Optimal Biomarker Sources: A Comparative Analysis of Exosomes and Whole Plasma in Fasting and Non-Fasting Conditions for Liquid Biopsy Applications"

_ijms, 2023, doi:10.3390/ijms25010371_

Round 1

Reviewer 1 Report

Comments and Suggestions for Authors

Summary of the Manuscript:

Reviewed the manuscript " Comparison of Liquid Biopsy Biomarker Sources: Exosome vs. Whole Plasma in Fasting and Non-Fasting States" by Masaki Nasu, et al. The study investigates the efficacy of exosomes and whole plasma as sources of biomarkers in liquid biopsies, specifically analyzing small non-coding RNAs (sncRNAs) and metabolites in fasting and non-fasting states. Key findings include that sncRNA profiles are consistent regardless of fasting, with exosomal sncRNAs presenting more consistent profiles than plasma. The study also observes notable concentration shifts in certain metabolites in whole plasma related to fasting status. The research concludes that fasting is not required for sncRNA and selected metabolomic profiling in liquid biopsies and advocates for the use of exosomal RNAs to mitigate batch effects due to hemolysis.

Strengths

  1. Innovative Approach and Comprehensive Analysis: The study's focus on sncRNAs and metabolites in various metabolic states is highly commendable.
  2. Methodological Transparency: The detailed methodology enhances the study's reproducibility.

Weaknesses

  1. Limited Sample Size: The study's conclusions are based on a small sample size, which may not accurately represent a diverse population.
  2. Incomplete Metabolite Analysis (Lines 534-546): The examination of metabolites, especially in exosomes, seems less exhaustive than the analysis of sncRNAs.
  3. Potential Bias in Data Interpretation (Lines 684-694): The manuscript appears to favor the superiority of exosomes without addressing their limitations comprehensively.
  4. Lack of Clinical Correlation: The manuscript does not sufficiently tie the findings to direct clinical applications or implications.

Recommendations

  1. A larger and more diverse sample cohort would enhance the generalizability of the findings.
  2. A more thorough investigation into metabolites, particularly those in exosomes, is needed.
  3. It would be beneficial to include pathological samples to assess the biomarkers' efficacy in disease settings.

Conclusion

This study is a valuable contribution to the field of liquid biopsy research. However, to enhance its impact, addressing the highlighted weaknesses is recommended.

Overall Recommendation: Minor Revision

The manuscript provides significant insights but would benefit from addressing the noted concerns, especially regarding sample size and comprehensive metabolite analysis.

Reviewer 2 Report

Comments and Suggestions for Authors

An interesting and rather well-written manuscript showing new methodological aspects of measuring exosomes in biological material. Below are my comments:

1. I ask the authors to consider changing the title of the manuscript to a more general one,

2. the basic question that I, as a reviewer, have is why plasma and not serum?

3. fonts of different sizes were used in the introduction,

4. Fig. 1 is cut off on one side and is of poor quality,

5. the first subsection in the results should be the characteristics of exosomes. The authors should examine the size of EVs and the expression of surface antigens typical of exosomes. How confident are the authors that they have examined exosomes?

6. moreover, the rest of the manuscript is prepared carefully,

7. the literature must be expanded with the following references:

a. https://www.mdpi.com/2073-4409/11/18/2913

b. https://www.mdpi.com/2075-1729/13/10/2033

c. https://www.mdpi.com/2072-6694/14/3/500

d. https://www.mdpi.com/1422-0067/23/17/9930

Round 2

Reviewer 2 Report

Comments and Suggestions for Authors

The authors have satisfactorily responded to all my questions and made the necessary changes to the manuscript.